# Development of a Sensitive and Specific Quantitative RT-qPCR Method for the Detection of Hepatitis E Virus Genotype 3 in Porcine Liver and Foodstuff

**DOI:** 10.3390/foods13030467

**Published:** 2024-02-01

**Authors:** Jan Bernd Hinrichs, Antonia Kreitlow, Madeleine Plötz, Ulrich Schotte, Paul Becher, Nele Gremmel, Roger Stephan, Nicole Kemper, Amir Abdulmawjood

**Affiliations:** 1Institute of Food Quality and Food Safety, University of Veterinary Medicine Hannover, 30173 Hannover, Germany; jan.bernd.hinrichs@tiho-hannover.de (J.B.H.); antonia.kreitlow@tiho-hannover.de (A.K.); madeleine.ploetz@tiho-hannover.de (M.P.); 2Department C Animal Health and Zoonoses, Central Institute of the Bundeswehr Medical Service Kiel, 24119 Kronshagen, Germany; ulrichschotte@bundeswehr.org; 3Institute of Virology, University of Veterinary Medicine Hannover, 30559 Hannover, Germany; paul.becher@tiho-hannover.de (P.B.); nele.gremmel@tiho-hannover.de (N.G.); 4Vetsuisse Faculty, Institute for Food Safety and Hygiene, University of Zurich, 8057 Zurich, Switzerland; stephanr@fsafety.uzh.ch; 5Institute for Animal Hygiene, Animal Welfare and Farm Animal Behaviors, University of Veterinary Medicine Hannover, 30173 Hannover, Germany; nicole.kemper@tiho-hannover.de

**Keywords:** hepatitis E virus genotype 3, RT-qPCR, pork products, food chain control, public health

## Abstract

As an international and zoonotic cause of hepatitis, hepatitis E virus (HEV) poses a significant risk to public health. However, the frequency of occurrence and the degree of contamination of food of animal origin require further research. The aim of this study was to develop and validate a highly sensitive quantitative RT-qPCR assay for the detection and quantification of HEV contamination in porcine liver and food. The focus was on genotype 3, which is most common as a food contaminant in developed countries and Europe. The selected assay has its target sequence in the open reading frame 1 (ORF1) of the HEV genome and showed good results in inclusivity testing, especially for HEV genotype 3. The developed assay seems to show high efficiency and a low intercept when compared to other assays, while having a comparable limit of detection (LOD). In addition, a standard curve was generated using artificially spiked liver to provide more accurate quantitative results for contamination assessment and tracking in this matrix. Application of the assay to test 67 pig livers from different origins resulted in a positivity rate of 7.5%, which is consistent with the results of numerous other prevalence studies. Quantitative detection of the viral genome in the food chain, particularly in pig livers, is essential for understanding the presence and evolution of HEV contamination and thus ensures consumer safety.

## 1. Introduction

Hepatitis E virus (HEV) is a globally occurring non-enveloped (+)-ssRNA virus that has been identified as the causative agent of human hepatitis E [1]. The virus belongs to the species *Paslahepevirus balayani* in the genus *Paslahepevirus* and within the family *Hepeviridae* and is currently divided into 8 different genotypes [2]. While genotypes 1 and 2 are mainly pathogenic for humans, genotypes 3 to 8 have been shown to infect other mammals with zoonotic potential for genotypes 3, 4, and 7 [3].

Transmission of the virus is mainly fecal–oral, with water being an important vector, especially for HEV genotypes 1 and 2, which are most common in developing countries.

In contrast, sporadic outbreaks in industrialized countries are mostly linked to the potentially zoonotic genotypes 3 and 4 [4,5].

Comparatively high anti-HEV-specific IgG prevalences indicate a wide distribution of the virus in Germany and Europe [6]. Although infection with genotype 3, which is the most common in Europe, usually appears to be subclinical, the health risk, especially for immunocompromised patients, cannot be neglected [5,7].

Since human-to-human transmission does not seem to play a major role with this genotype, zoonotic infections are assumed to play a relevant role in the spread of HEV in Europe [8]. The HEV genotype 3 primarily infects pigs in addition to humans, and the European domestic pig population, as well as wild boar, is suspected to act as an important virus reservoir [9,10].

Higher antibody prevalence in occupational groups with frequent contact with pigs suggests that direct transmission of the virus from pigs to humans is possible [11,12,13]. However, in various outbreaks of hepatitis E, a clear phylogenetic proximity of the virus detected in the patients to the HEV found in the food suspected as the source of infection could also be demonstrated [8,14]. It is therefore assumed that transmission through food of animal origin, especially from pigs, also plays a significant role in the spread of hepatitis E in Europe. In addition to the high antibody and virus prevalence in the European domestic and wild boar population, HEV RNA has been found in a large number of studies at various stages of the food chain, both in pork liver at the slaughterhouse as well as in ready-to-eat sausages in shops [15,16,17,18].

Since the virus appears to replicate mainly in hepatocytes, foods containing pork liver are particularly suspected as a source of infection [19]. Several outbreaks in Europe attributable to foodstuff were also linked to the consumption of insufficiently heated pork liver and liver sausages containing pork liver [8,20,21]. Therefore, meat from pigs or wild boar and pork liver sausage have been identified as risk factors for HEV infection in various countries, as well as oysters and offal [22,23]. As a result, there have been increasing proposals in recent years to improve the monitoring of food of animal origin and its raw materials for HEV contamination.

A better understanding of the development of HEV contamination during the processing of foods containing potentially contaminated parts, and in particular food containing pig liver, would also be essential to improve consumer safety [24].

As sufficiently effective cell lines for the cultivation of HEV have only recently become available and there is still no standardized procedure for the multiplication of the virus, as well as serological detection methods not being considered sufficiently sensitive, the detection methods are mostly based on the qualitative detection of the HEV genome [25,26].

With the official testing procedure BVL L 06.17.01-1:2020-11, a standardized and proven method for the extraction and qualitative detection of HEV in food is available in Germany, which uses the assay of Jothikumar et al. 2006 [27] to detect the virus RNA.

However, particularly for the assessment of the contamination level in the food chain and its distribution as well as its development, very sensitive quantitative detection is needed to assess the impact of individual measures, such as improved hygiene. A standard method for the quantification of HEV in food has not yet been established, which makes it difficult to assess the level of contamination in the food chain as accurately as possible.

This problem is partly due to the fact that most of the assays developed to date have focused on a sufficiently inclusive detection of HEV. This is made difficult by the high genetic diversity of the virus, which is particularly pronounced for genotype 3, the most relevant in food in Europe. Most of these assays target ORF2 or the partially overlapping ORF3 of the HEV genome [28]. The ORF3 region is the most homogeneous within the genome and lends itself to very inclusive detection of all genotypes [29]. However, other regions of the HEV genome may also be suitable for more sensitive detection of the food-relevant genotypes, such as a region at the beginning of ORF1 used by Mizuo et al. [30] or another region in ORF2 used by Erker et al. [31]. Most of the detection methods developed so far and applied to food are RT-qPCR assays [28], but some LAMP assays have also been presented in East Asia for monitoring contamination, especially with genotype 4 [32,33,34]. However, due to the highly heterogeneous HEV genome and the resulting difficulty in selecting a large homologous target area, especially in genotype 3, the RT-PCR assay seems to be more suitable for detection.

Most PCR assays developed to date were originally developed for the detection of HEV in serum from clinically ill patients or the detection in water, but only a few of these assays have been validated or developed for the detection of HEV contamination in food [28].

An example of a commonly used method is the PCR assay for water testing developed by Jothikumar et al., in 2006 [27], which was also tested with HEV-contaminated samples from pigs.

This assay has been widely and successfully used for the qualitative detection of HEV in the food chain and is characterized by its high inclusivity and the associated low probability of false-negative results [35]. It has also been used for quantification after calibration with an RNA-standard. However, especially for the quantitative evaluation of results from food samples, this test does not show optimal results due to the comparatively high ct values for genotype 3 samples.

Highly sensitive detection is required for the optimal quantitative interpretation of results, but this may compromise the inclusivity. For the quantitative evaluation of HEV in the blood of human patients, great progress has been made in recent years with the introduction of an international standard [36] for the calibration of the detection method, so that several quantitative detection methods are already available, e.g., Germer et al. [37] or Frias et al. [38]. For the highly heterogeneous and difficult to process food matrices, there is no such standardized material suitable for calibration, making the development of a quantitative detection method much more difficult.

The lack of such a standard material makes it necessary to test the application of a potentially quantitative method with either natively contaminated liver or appropriately spiked liver. In the case of naturally contaminated liver, sufficient extraction of HEV RNA from the matrix must be ensured, which is usually performed using an internal extraction control. Examples of such controls are the *E. coli* phage (MS2) or the *murine norovirus* (MNV-1), which are usually detected simultaneously with HEV detection in a multiplex assay [35].

One problem for spiking liver was the difficulty of specifically multiplying the HEV in cell culture, making it hard to obtain quantified material for such spiking. A uniform approach in which the sample used for spiking can be used both as an extraction control and to calibrate the quantification of the detection of HEV RNA in complex food matrices would be optimal, so that a calibration covering all steps is possible.

The aim of this study was to develop and validate a highly sensitive quantitative RT-qPCR method for the detection and quantification of HEV genotype 3 contamination in pig liver and food, considering the relevant HEV subtypes and taking into account losses during isolation and transcription in the calibration process.

## 2. Materials and Methods

### 2.1. Primer Design

Primers and probes were designed using the Primer-BLAST online design tool [https://www.ncbi.nlm.nih.gov/tools/primer-blast/, accessed on 17 February 2023] provided by the National Center for Biotechnology Information (NCBI) (Bethesda, MD, USA). The primers and probes were designed to be homologous to the reference sequences proposed by Smith et al., (2020) [2] of five different subtypes of HEV genotype 3 frequently found in food of animal origin in Europe [15,17]. Sequence data was obtained from the GenBank provided by NCBI [https://www.ncbi.nlm.nih.gov/genbank/, accessed on 17 February 2023] and included the reference genomes of HEV genotypes 3c, 3f, 3b, 3e, and 3i with GenBank accession numbers FJ70535, AB369687, AP003430, AB248521, and FJ998008, respectively. To prevent cross-reactions with other viruses belonging to the *Norovirus* and *Hepatovirus* genera, potential matches within the amplicon were excluded. This was achieved through computational alignment of the target sequence with the genomes of these viruses, available in the GenBank database.

On this basis, five different primer sets targeting regions in the ORF1 and ORF3 regions of the HEV genome were selected and pre-tested using a SYBR Green Real-time PCR to detect the first international HEV virus panel provided by the Paul Ehrlich Institute (PEI) (Langen, Germany) and a variety of HEV eluates from field samples. For the pre-test, HEV RNA was converted into DNA using a QuantiTect Reverse Transcription Kit™ (Qiagen GmbH, Hilden, Germany) and then analyzed using the FastStart Essential DNA Green I Mastermix™ and a LightCycler^®^ 96 device from Roche Diagnostics GmbH (Mannheim, Germany). A complementary probe was designed for the primers that showed the best inclusivity and detection speed in these pre-tests, and this primer set was then used for further testing in a TaqMan PCR assay. The selected primer sequences (See Table 1) targeted an amplicon in the ORF1 region of the HEV genome located between bp 45 and 131 in the reference strain HEV subtype 3c (FJ705359). All primers and probes were ordered from *Eurofins* Genomics Germany GmbH (Ebersberg, Germany).

### 2.2. Implementation of an Internal Amplification Contral (IAC)

An IAC was implemented to assess the possible inhibition of the reaction by background DNA and other substances such as fat. To achieve this, the primers and probes proposed by Anderson et al., (2011) [39], which have been used successfully in other food-optimized detection assays were used (See Table 1). These primers target a sequence in the tobacco (*Nicotiana tabacum*) genome. A DNA-oligonucleotide matching this primer set’s target sequence was prepared and quantified by Eurofins Genomics Germany GmbH and added to each reaction. To determine the optimal concentration of the amplification control in each reaction, a dilution series of the oligonucleotide was prepared at concentrations between 1 pmol/µL × 10^−5^ and 1 pmol/µL × 10^−12^ and tested in several runs using the PCR settings described in Section 2.3. Potential inhibition of the HEV RNA detection reaction by the IAC or the added primer set was also evaluated during this trial, by adding an HEV positive control to each reaction at a concentration close to the detection limit. The optimum concentration for the IAC, at which the IAC was close to its detection limit and no inhibition of HEV detection was observed during the preliminary testing, was determined to be 1 pmol/µL × 10^−11^, and 1 µL of this was added as IAC to each reaction.

### 2.3. Reverse Transcription Quantitative Real-Time PCR (RT-qPCR) Assay

The RT-qPCR was performed using 10 µL 2× master mix (Takyon™ No ROX 2X MasterMix), 0.2 µL reverse transcriptase and 0.2 µL additives provided by Takyon™ to stabilize the reverse transcription. The reagents were produced and provided by Eurogentec (Seraing, Belgium). The concentration of primers for the detection of HEV-RNA was 250 nM and 100 nM for the associated probe. For the implemented IAC, primer and probe concentration was 500 nM and 200 nM, respectively. The primer sets used are found in Table 1. Nuclease-free water was added to bring the total reaction volume to 20 µL. Reverse transcription was performed at a temperature of 48 °C for 20 min, followed by an initial denaturation step at 95 °C for 3 min. Subsequently, forty cycles of a two-step amplification were carried out, consisting of denaturation at 95 °C for 10 s and annealing at 60 °C for 60 s. At the end of each amplification cycle, amplification was detected by measuring an increase of fluorescence using the LightCycler^®^ 96 device (Roche Diagnostics GmbH). The results were recorded and analyzed using the Light Cycler^®^ Application Software 1.1 (Roche Diagnostics GmbH, Mannheim, Germany).

For each reaction, a positive control was performed in one reaction vessel to ensure a successful reaction, while a negative control was performed in another vessel to confirm that no non-specific reactions occurred.

### 2.4. Reference Method Used for Comparative Analysis of Samples

The official standard method for the qualitative detection of HEV in swine liver in accordance with BVL L 06.17.01-1:2020-11 was selected as reference method. Slight modifications were made to the RNA extraction and the PCR reaction parameters. For RNA extraction, a larger amount of tissue was first minced with a scalpel and mixed, of which 0.25 g of tissue was then used for the first homogenization step. This reduction was conducted in order to achieve a higher cell disruption in accordance with recent studies [40]. The reaction parameters of the PCR were adjusted accordingly in order to achieve optimum adaptation to the PCR device and the reagents used. Reverse transcription was therefore carried out at a temperature of 48 °C as recommended by the manufacturer of the reverse transcriptase (Takyon™). In addition, a shorter initial denaturation of 3 min at 95 °C was chosen, also as recommended by the manufacturer of the master mix, Takyon™. Amplification was performed for 45 cycles as a two-step amplification [Temp. °C, Ramp. °C/s (Duration s); 95 °C, 4.4 °C/s (10 s); 60 °C, 2.2 °C/s (30 s)] as recommended by the manufacturer of the PCR instrument (LightCycler^®^ 96, Roche Diagnostics GmbH, Vienna, Austria).

### 2.5. Analytical Specificity of the JBH4-HEV RT-qPCR Assay

A total of 46 samples containing HEV RNA from different subtypes of HEV genotypes 3 and 4 were tested to determine the inclusivity of the JBH4-HEV RT-qPCR assay. The sample eluates tested were HEV reference material from the Paul Ehrlich Institute [36], RNA isolated from wild boar liver in previous HEV prevalence studies by Schotte et al. [41], RNA isolated from liver samples from the study by Wist et al. [42], and RNA isolated from field samples of wild boar and domestic pig liver by Gremmel et al. [43]. The selected subtypes were primarily chosen due to their association with foodborne transmission and their significance for human infections, covering HEV subtypes 3c, 3f, 3i, 3e, 3h, 3-i-like, 3b, 3ra, 4c and 4g [44]. Additionally, the exclusivity of the JBH4-HEV RT-qPCR assay was investigated using non-target virus material from closely related species and viruses circulating in the same environment as HEV. The tests included DNA and RNA of various enveloped and non-enveloped viruses provided by the Institute of Virology of the University of Veterinary Medicine Hannover. Potential cross-reactivity was screened with *Porcine circovirus 3* (PCV3), *Porcine parvovirus* (PPV), *Swine influenza virus* (SIV), *Pseudorabies virus* (PRV), *Atypical porcine pestivirus* (APPV), *Transmissible gastroenteritis virus* (TGEV) and the EU and US variants of the *Porcine reproductive and respiratory syndrome virus* (PRRSV). Cross-reactivity with other food-borne viruses such as norovirus or hepatitis A virus was also assessed by digital alignment of the amplicon with known sequences of these viruses available in the GenBank database.

In addition, RNA was isolated from muscle, fat, and liver tissues from pigs previously found to be HEV-negative by both the new assay and the reference method and tested to exclude possible matrix-related cross-reactivity. All eluates were screened using the RT-qPCR presented in this study and the previously described reference RT-qPCR to further validate the results.

### 2.6. Analytical Sensitivity of the JBH4-HEV RT-qPCR Assay

Analytical sensitivity was first determined using an RNA oligonucleotide designed to be complementary to the target region of the JBH4 HEV primer set in the ORF1 region of HEV subtype 3c (accession number FJ705359). The RNA oligonucleotide was generated and quantified by Eurofins Genomics GmbH. The specified stock concentration of 100 pmol/µL was first adjusted to 1 genome equivalent (GE)/µL × 10^13^. A dilution series from 1 × 10^7^ GE/µL to 1 × 10^0^ GE/µL was then prepared and tested in 6 replicates. Each dilution step was added directly to the reaction mixture and tested with the newly developed PCR assay as described above. The data obtained during the investigation of this dilution series under minimal external influences was used to create an initial standard curve.

This standard curve was used to assess possible losses during the isolation steps in the spiking experiment and to evaluate these overall.

### 2.7. Quantification of HEV RNA by the JBH4-HEV RT-qPCR Assay under Field Conditions by a Spiking Experiment

In order to quantify the samples tested as realistically as possible, an additional standard curve was generated based on a spiking experiment. By comparing this curve with the previously generated analytical standard curve based on the pure RNA oligonucleotide, it was possible to account for both isolation losses and interfering background substances from the liver sample.

For this purpose, 6 portions of 0.25 g liver tissue each were taken from a liver that had previously tested negative with both the new PCR and the reference PCR. After initial homogenization, three 25 mg portions were taken from each of these six samples and individually transferred to a new tube containing 600 µL RLT buffer (without ß-mercaptoethanol/DTT) and a GK60 Precellys Lysing Kit™ (Bertin-Technologies, Montigny-le-Bretonneux, France). Prior to the second homogenization step, each sample was spiked with 10 µL of the RNA oligonucleotide at a specific dilution. The concentrations used ranged from 1 × 10^7^ GE/µL to 1 × 10^0^ GE/µL and were each tested three times, giving a total of nine samples. After the second homogenization step, the samples were further processed according to the protocol of the Qiagen RNeasy kit™ (Qiagen GmbH) and the isolated RNA was eluted in 50 µL of RNase-free water. If the spiked RNA was completely isolated, the resulting eluates would concentrations are to be expected that are approximately between 1 × 10^7^ GE/µL and 1 × 10^0^ GE/µL. The limit of detection was set at the dilution level at which 2/3 of the samples could be detected.

Treatment was performed on ice to minimize potential losses and isolated RNA eluate was immediately placed on ice and analyzed directly by RT-qPCR.

### 2.8. Validation of the JBH4-HEV RT-qPCR Assay by Testing Naturally Contaminated Porcine Liver

To evaluate the suitability of the assay for testing field samples, RNA was extracted from 67 livers and tested for HEV RNA using both the reference PCR method [27] and the newly established JBH4-HEV RT-qPCR assay. The livers were randomly collected from slaughterhouses, hunters, and retailers located in the vicinity of Hannover city. Liver samples were processed using a slightly modified extraction protocol according to the official method BVL L 06.17.01-1:2020-11. Modifications were made according to the publication by Zhao et al. [40]. Also longer centrifugation times were chosen.

Briefly, approximately 50 g of liver tissue was collected from four different subserosal sites, pre-mixed, weighed to a total weight of 0.25 g and then placed in a 2 mL reaction tube filled with zirconium beads and 250 µL of 1% phosphate-buffered saline. The sample was then homogenized using the Precellys-Evolution™ homogenizer (Bertin-Technologies) for 5 cycles of 30 s each at a speed of 5 ms^−1^. The sample was then centrifuged at 13,000× *g* for 20 min at 5 °C and then 25 mg of the supernatant homogenate was added to a new tube containing 600 µL of buffer RLT (Qiagen GmbH) and homogenized again using the GK60 Precellys Lysing Kit™ (Bertin-Technologies) for 1 cycle of 20 s at a speed of 5 ms^−1^. The RNA was then extracted and enriched by ultrafiltration using the Qiagen RNeasy Kit™ (Qiagen GmbH) according to the manufacturer’s recommendations. All RNA samples were analyzed using both the newly developed assay and the established reference method to verify the results.

Quantitative evaluation of HEV contamination in samples tested positive was estimated using the standard curve generated within the previously described spiking experiment.

## 3. Results

### 3.1. Analytical Specificity of the JBH4-HEV RT-qPCR Assay

The JBH4-HEV RT-qPCR assay was able to detect all genotype three samples tested (see Figure 1), while the two genotype four samples showed no measurable amplification (see Table 2). Of the 46 HEV eluates tested, three were negative in the reference PCR, two of which belonged to genotype 3i and one to genotype 4g. The RNA samples isolated from liver and muscle of pigs previously tested negative for HEV in the reference assays also showed no detectable amplification within 40 amplification cycles. None of the samples from a total of eight non-target viruses showed a positive reaction within the 40 amplification cycles, while the IAC was detected in each reaction and showed no evidence of inhibition. There was also no detectable match between PCR primers and genome sequences of the Norovirus and Hepatovirus genera available in NCBI GenBank using the NCBI BLAST algorithm for alignment.

### 3.2. Analytical Sensitivity of the JBH4-HEV RT-qPCR Assay

In the first experiment, the target RNA was detected up to a dilution level of 1 × 10^1^ GE/µL (GenomeEquivalents/µL), corresponding to a number of 20 copies per reaction.

Results are found in Table 3. From the six-fold repetition of the measurement of the individual concentration levels, a standard curve was calculated, which could be described with the equation y = −3.295x + 37.811. The value m (−3.295) indicated the slope of the curve and the value b (37.811) indicated the Y-value for log level 0, i.e., the calculated Cq-value for the detection of 1 copy per reaction. Using the formula E = (10^(−1/slope)^)^−1^, the efficiency was then calculated from its correlation with the slope. The equation E = (10^(−1/−3.1216)^)^−1^ thus resulted in a PCR efficiency of 1.011. R^2^ was determined at 0.9909.

### 3.3. Analytical Sensitivity of the JBH4-HEV RT-qPCR Assay under Field Conditions in Spiked Liver Samples

Subsequently, the target RNA was also used to spike liver material for generating a matrix-specific standard curve. As described in Section 2.7, 10 µL of each of the dilution levels 10^6^ to 10^1^ GE/µL were spiked, so that a total of 10^7^ to 10^2^ GE were present in 25 mg of material. The corresponding results are shown in Table 4.

The resulting standard curve could be described with the equation y = −3.3297x + 38.569. The equation E = (10^(−1/−3.3297)^)^−1^ resulted in a PCR efficiency of 0.9967, and R^2^ at 0.9888 (see Figure 2). The detection limit was set at 1 × 10^3^ GE/25 mg liver material. No reaction was affected by inhibition as the IAC was in every sample.

### 3.4. Validation of the JBH4-HEV RT-qPCR Assay

Of the 67 livers tested, a total of 5 showed a positive reaction in both the new assay and the reference method BVL L 06.17.01-1:2020-11. Of these 5 positive livers, 4/5 were from domestic pigs and 1/5 from wild boar, 3/5 were purchased from local butcher shops, 1/5 was purchased directly at the slaughterhouse and one was from hunting. The contaminations were quantified and fluctuated considerably with values between 972 GE/25 mg and 215,583 GE/25 mg.

## 4. Discussion

The molecular detection of food contaminants using PCR is challenging due to the high heterogeneity of the matrices and the associated potentially high proportion of interfering substances. The high genetic diversity of the HEV genome, particularly genotype 3, which is one of the predominant food contaminants in Europe, makes it difficult to select a suitable target region and specific primers for detecting the presence of the HEV genome [2,29,46].

Inoue et al. [47] identified three regions in the HEV genome that could be suitable for inclusive molecular detection of HEV due to a sufficiently high sequence identity of over 75%. These include the 5′-untranslated-region (UTR) with the 5′-terminal part of ORF1, the overlapping ORF2/ORF3 region and the central part of ORF2. Most assays focus on the regions in ORF2 and ORF3 to achieve high inclusivity, such as the most commonly used assay for the detection of HEV in food by Jothikumar et al. [27]. In the case of a prioritized consideration of HEV genotype 3 subtypes, and thus the major HEV genotype in pork-containing foods, primer design in ORF1 is sufficiently comprehensive and may offer advantages in terms of sensitivity. In this study, various primer sets were tested in preliminary tests. The selected ORF1-based primer set outperformed the alternative variants in terms of analytical inclusivity and sensitivity.

Since most of the developed assays to date were originally validated for human diagnostics and were only partially used for HEV detection in food, comparable data about their performance in food analysis are lacking. An exception is the HEV assay by Jothikumar et al. [27], which has also been established for investigating environmental samples. As this assay has been shown to be probably the most sensitive for detecting HEV in pig samples by Ward et al. [35], this assay was used in many prevalence studies including food samples [28]. For this reason, it was selected as reference assay in this study. A total of 46 HEV RNA containing samples were analyzed with both the new JBH4_HEV RT-qPCR assay and the reference assay. The three previously confirmed HEV genotype 3 samples that tested negative using the assay by Jothikumar et al. [27] could be correctly identified by the new JBH4_HEV RT-qPCR suggesting a higher sensitivity. In addition, inclusivity was tested using samples of HEV genotypes 3 and 4 from the World Health Organization’s first international reference panel for different hepatitis E virus genotypes for nucleic acid amplification assays [36]. All HEV genotype 3 samples were tested positive whereas the 2 samples of genotype 4, the predominant food-associated HEV variant in East Asia, were not detectable using the JBH4_HEV RT-qPCR. The genotype 4g sample was tested negative in both the new and reference PCR assay while genotype 4c was amplified with a comparatively high cq value of 38 in the reference PCR assay. Insufficient RNA isolation could be considered as the cause for the false negative results, as the selected isolation method was not optimized for serum samples.

The inclusivity test performed in this study focused on subtypes of HEV genotype 3, which are most frequently detected in European foods. As most PCR assays are designed to detect the different HEV genotypes associated with humans, they have been validated for a wide range of genotypes, but with less emphasis on HEV genotype 3 in and from food. For example, Enouf et al. [40] tested eight HEV genotype 3 eluates from patient serum, whereas many other assays determined inclusivity only by comparison with genomic databases. In the study of Jothikumar et al. [27], only 10 HEV genotype 3 eluates were considered. By testing 42 different HEV eluates, the inclusivity of the JBH4_HEV RT-qPCR assay was therefore more thoroughly validated for food-related applications.

Investigating the exclusivity of the JBH4_HEV RT-qPCR included viral contaminants that potentially occur in porcine liver. Some PCR assays developed for detection of HEV showed a similar strategy in their exclusivity testing, e.g., Kaba et al. [48] and Jothikumar et al. [27]. With no false positives, the JBH4_HEV RT-qPCR performed similarly to previously established assays.

The determination of analytical sensitivity and the development of a quantitative HEV-specific RT-qPCR assay is a major challenge, as until recently there were no efficient cell culture models for virus propagation and subsequent quantification of the purified analyte [26]. Although precise quantification of, e.g., liver material is already possible with digital droplet PCR [49], this method is not yet suitable for mass production due to its limited distribution and high costs. Quantified reference material, such as the International Reference Panel [36], which is available for the evaluation of PCR tests for human diagnostics based on serum or fecal samples, does not exist in this form for pig liver and therefore for food matrices. In order to avoid the need for virus replication, artificially produced DNA- and RNA-oligonucleotides, which are identical to the amplicon, can be used for determining analytical sensitivity and detection limit in various food products and raw materials.

This strategy can help to achieve a good comparability between the performance of different assays. For example, compared with the four standard curves obtained by Ward et al. [35] using various HEV-specific TaqMan-RT-qPCR assays, the JBH4_HEV RT-qPCR assay showed a very good efficiency of 101.1% and a comparatively low intercept of 37.81 for the pure analyte. A lower intercept value of 35.96 was only achieved by Ahn et al. [50]. However, their assay showed limitations in terms of inclusivity for the HEV genotype 3 occurring in pigs.

The analytical sensitivity determined for the JBH4_HEV RT-qPCR assay ranged between 2 and 20 GE per reaction and thus, was comparable to PCR assays developed by Gyarmati et al. [51] (1 and 20 GE per reaction), Enouf et al. [52] (10 GE per reaction), Ahn et al. [50] (16.8 GE per reaction) and Jothikumar et al. [27] (4 GE per reaction). In this study, these data were supplemented by a spiking experiment using the amplicon-based RNA oligonucleotide for artificial contamination of porcine liver to assess the detection limit and performance of the assay under the influence of matrix-specific factors.

The achieved detection limit of 20 GE/reaction, an efficiency of 99.67% and a relatively low intercept of 38.57 for the standard curve obtained confirmed that the JBH4_HEV RT-qPCR assay was suitable, highly sensitive and reliable tool for detecting and quantifying HEV in complex samples even taking into account potential RNA losses during sample processing. The cq values measured for each sample were reproducible in all repetitions and showed an overall average deviation of 0.69 Ct. It can be concluded that the RNA yields were stable after each isolation process and that the extraction method used was compatible with the subsequent JBH4_HEV RT-qPCR assay and sufficiently robust. This evaluation of the extraction method including mechanical disruption and subsequent ultrafiltration was consistent with the findings of Hennechart–Collette et al. [53] who recommended the procedure for use on liver sausages.

When using the JBH4_HEV RT-qPCR assay to analyze pig livers from various sources in the Hannover region, 5 out of 67 livers were found to be positive, which corresponds to a prevalence of 7.5%. This result is therefore very close to the results of various prevalence studies on pig livers in Germany and Europe [9], such as the prevalence of 4% found by Wenzel et al. [54] for pig livers from domestic pigs in south-western Germany or the prevalence of 13.5% and 11% found by Baechlein et al. [55] and Boxmann et al. [56] for pig livers from domestic pigs in Germany and the Netherlands, respectively.

Comparably high percentages of positive liver samples were also found in HEV prevalence studies in wild boar, such as the 5.9% positive samples found by Gremmel et al. [43] in northern Germany or the values of 2.8% to 13.3% HEV-positive samples between 2013 and 2017 described by Schotte et al. [41] in various regions of Germany.

The difference in prevalence in the present study may be due to the smaller sample size of 67 livers, considering that Gremmel et al. [43], Baechlein et al. [55] and Boxmann et al. [56] sampled over 200 animals, while Schotte et al., sampled approximately 3500 [41].

In the study by Boxmann et al. [56], the average contamination of the livers analyzed was log(10) 3.2 GE/0.1 g with a range between 1.7–6.2 GE; in the study by Baechlein et al. [55], an average load of log(10) 8 GE/0.1 g is stated.

This range of results is therefore largely consistent with the results of this study, in which an average native viral load of log(10) 4.8 GE/0.1 g was determined in pig liver, with a range of 3.8 to 6.11 GE/0.1 g. In most studies, however, the HEV load of the samples was not quantitatively assessed, so that only very limited comparative data are available [48,57,58].

In general, it is difficult to estimate the risk for consumers from exposure to contaminated pig liver and products thereof. This is due to the fact that no human infectious dose has yet been established for HEV, so reference values from pigs or monkeys have to be used [25]. Furthermore, only limited conclusions can be drawn about the infectivity of the virus from positive PCR results because the method only detects RNA, but not the extent to which it is also a component of an infectious virus or an already partially denatured structure. Previously proposed methods of differentiation, such as the addition of RNase to denature free viral RNA or selective dyes, can only be indicative [59]. A recent study focusing on the isolation of HEV from liver samples showed that 83.3% (15 out of 18) HEV RNA-positive samples contained infectious hepatitis E viral particles and therefore must be considered as a potential source for human infections [43]. Overall, the JBH4_HEV RT-qPCR assay could be adapted as a good tool for use in relevant food matrices with an increased risk of HEV contamination, such as meat products containing pork liver. Specializing and optimizing the assay for these matrices could thus help to track the progression of HEV contamination in the food chain by quantifying the contamination within the different production stages, improve food control, and verify the effectiveness of reduction measures. It can therefore make a significant contribution to protecting consumers from HEV infection by improving food safety through better identification of critical control points in the processing of HEV-contaminated pork liver and through sensitive monitoring of potentially contaminated food. Further research is required to investigate the occurrence and development of the hepatitis E virus in the European food chain [24], including substantial screening of animal food stuffs, particularly pig liver, and more precise recording of the possible food-related spread. Finally, the JBH4_HEV RT-qPCR assay is a suitable tool for generating comprehensive and detailed data to fill these gaps.

## Figures and Tables

**Figure 1 foods-13-00467-f001:**
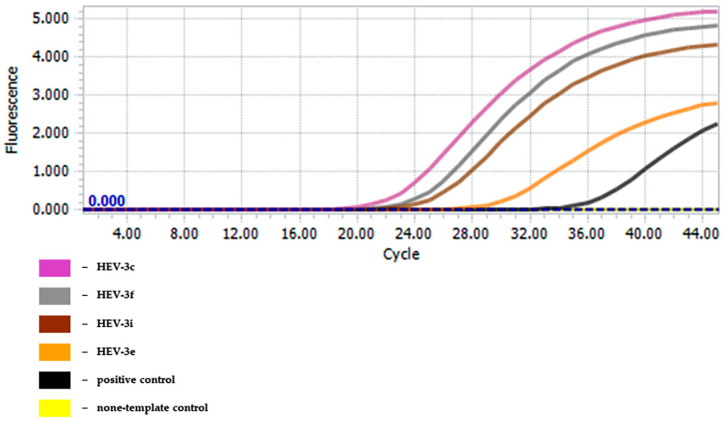
Amplification curves from the analysis of different HEV-positive samples isolated from pig liver and assigned to the HEV genotype 3.

**Figure 2 foods-13-00467-f002:**
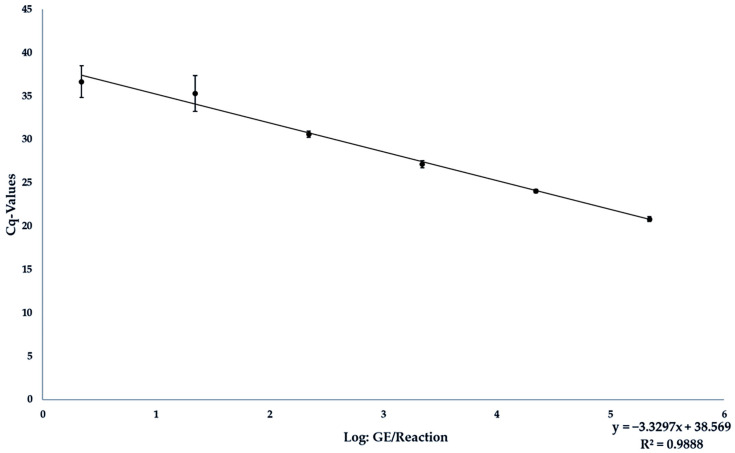
Standard curve from the results of measuring the analytical sensitivity of the JBH4-HEV RT-qPCR assay under field conditions by analyzing liver spiked with different dilution levels of target RNA.

**Table 1 foods-13-00467-t001:** Primers and probes used for RT-qPCR.

Primer:	Sequence 5′ → 3′	Region	GC%	Tm (°C)
**JBH4-HEV**		ORF1 (FJ705359)		
ForwardPrimer: Fw_JBH4-HEV	5′-TAAGGCTCCTGGCATTACTACT-3′	45–67	45.45	58.35
ReversePrimer: Rv_JBH4-HEV	5′-GCCGAACCACCACAGCATT-3′	113–131	57.89	61.27
Probe: P_JBH4-HEV	5′-[FAM]-CTGCTCTGGCTGCGGCCAA-[BHQ1]-3′	81–99	68.42	59.96
**IPC-ntb2**	Source: Anderson et al., (2011) [39]			
ForwardPrimer: IPC-ntb2-fw	5′-ACCACAATGCCAGAGTGACAAC-3′		50	68
ReversePrimer: IPC-ntb2-re	5′-TACCTGGTCTCCAGCTTTCAGTT-3′		47.82	68
Probe: IPC-ntb2	5′-[HEX]-CACGCGCATGAAGTTAGGGGACCA-[BHQ1]-3′		58.3	74

**Table 2 foods-13-00467-t002:** Viral eluates used for testing the analytical specificity of the JBH4-HEV RT-qPCR assay.

Virus	Origin	Reactivity
Inclusivity		
HEV-Subtype 3b	HEV reference material PEI (Baylis et al., 2012) [45]	+
HEV-Subtype 3c	HEV reference material PEI (Baylis et al., 2012) [45]	+
HEV-Subtype 3c	Eluate isolated from pigliver (Schotte et al., 2022) [41]	+
HEV-Subtype 3c	Eluate isolated from pigliver (Schotte et al., 2022) [41]	+
HEV-Subtype 3c	Eluate isolated from pigliver (Schotte et al., 2022) [41]	+
HEV-Subtype 3c	Eluate isolated from pigliver (Schotte et al., 2022) [41]	+
HEV-Subtype 3c	Eluate isolated from pigliver (Schotte et al., 2022) [41]	+
HEV-Subtype 3c	Eluate isolated from pigliver (Schotte et al., 2022) [41]	+
HEV-Subtype 3c	Eluate isolated from pigliver (Schotte et al., 2022) [41]	+
HEV-Subtype 3c	Eluate isolated from pigliver (Schotte et al., 2022) [41]	+
HEV-Subtype 3c	Eluate isolated from pigliver (Schotte et al., 2022) [41]	+
HEV-Subtype 3c	Eluate isolated from pigliver (Schotte et al., 2022) [41]	+
HEV-Subtype 3c	Eluate isolated from pigliver (Schotte et al., 2022) [41]	+
HEV-Subtype 3c	Eluate isolated from pigliver (Schotte et al., 2022) [41]	+
HEV-Subtype 3c	Eluate isolated from pigliver (Schotte et al., 2022) [41]	+
HEV-Subtype 3c	Eluate isolated from pigliver (Schotte et al., 2022) [41]	+
HEV-Subtype 3c	Eluate isolated from pigliver (Schotte et al., 2022) [41]	+
HEV-Subtype 3c	Eluate isolated from pigliver (Schotte et al., 2022) [41]	+
HEV-Subtype 3c	Eluate isolated from pigliver (Gremmel et al., 2022) [43]	+
HEV-Subtype 3c	Eluate isolated from pigliver (Gremmel et al., 2022) [43]	+
HEV-Subtype 3c	Eluate isolated from pigliver (Gremmel et al., 2022) [43]	+
HEV-Subtype 3c	Eluate isolated from pigliver (Gremmel et al., 2022) [43]	+
HEV-Subtype 3e	HEV reference material PEI (Baylis et al., 2012) [45]	+
HEV-Subtype 3e	Eluate isolated from pigliver (Gremmel et al., 2022) [43]	+
HEV-Subtype 3f	HEV reference material PEI (Baylis et al., 2012) [45]	+
HEV-Subtype 3f	Eluate isolated from pigliver (Schotte et al., 2022) [41]	+
HEV-Subtype 3f	Eluate isolated from pigliver (Schotte et al., 2022) [41]	+
HEV-Subtype 3f	Eluate isolated from pigliver (Schotte et al., 2022) [41]	+
HEV-Subtype 3f	Eluate isolated from pigliver (Schotte et al., 2022) [41]	+
HEV-Subtype 3f	Eluate isolated from pigliver (Schotte et al., 2022) [41]	+
HEV-Subtype 3f	Eluate isolated from pigliver (Schotte et al., 2022) [41]	+
HEV-Subtype 3f	Eluate isolated from pigliver (Schotte et al., 2022) [41]	+
HEV-Subtype 3h	Eluate isolated from pigliver (Wist et al., 2018) [42]	+
HEV-Subtype 3i	Eluate isolated from pigliver (Schotte et al., 2022) [41]	+
HEV-Subtype 3i	Eluate isolated from pigliver (Schotte et al., 2022) [41]	+
HEV-Subtype 3i	Eluate isolated from pigliver (Schotte et al., 2022) [41]	+
HEV-Subtype 3i	Eluate isolated from pigliver (Schotte et al., 2022) [41]	+
HEV-Subtype 3i	Eluate isolated from pigliver (Schotte et al., 2022) [41]	+
HEV-Subtype 3i	Eluate isolated from pigliver (Schotte et al., 2022) [41]	+
HEV-Subtype 3i	Eluate isolated from pigliver (Schotte et al., 2022) [41]	+
HEV-Subtype 3i-like	Eluate isolated from pigliver (Schotte et al., 2022) [41]	+
HEV-Subtype 3i-like	Eluate isolated from pigliver (Schotte et al., 2022) [41]	+
HEV-Subtype 3i-like	Eluate isolated from pigliver (Schotte et al., 2022) [41]	+
HEV-Subtype 3ra	HEV reference material PEI (Baylis et al., 2012) [45]	+
HEV-Subtype 4c	HEV reference material PEI (Baylis et al., 2012) [45]	-
HEV-Subtype 4g	HEV reference material PEI (Baylis et al., 2012) [45]	-
Exclusivity		
PCV3	IoV ^1^	-
PPV	IoV ^1^	-
SIV	IoV ^1^	-
PRV	IoV ^1^	-
APPV	IoV ^1^	-
TGEV	IoV ^1^	-
PRRSV EU	IoV ^1^	-
PRRSV US	IoV ^1^	-

^1^ Institute of Virology, University of Veterinary Medicine Hannover.

**Table 3 foods-13-00467-t003:** Detection of a dilution series of pure RNA oligonucleotide.

GE/µL	Positive/Tested	Average Cq-Value	Average Deviation
1 × 10^7^	6/6	13.24	0.217
1 × 10^6^	6/6	16.90	0.168
1 × 10^5^	6/6	20.46	0.059
1 × 10^4^	6/6	23.84	1.193
1 × 10^3^	6/6	26.92	0.132
1 × 10^2^	6/6	31.62	1.995
1 × 10^1^	6/6	33.83	0.880
1 × 10^0^	2/6	35.47	0.311

**Table 4 foods-13-00467-t004:** Detection rate of target RNA in artificially contaminated liver tissue at different contamination levels.

GE/25 mg Sample	Tested/Positive	Average Cq-Value	Average Deviation
1 × 10^7^	9/9	20.805	0.285
1 × 10^6^	9/9	24.058	0.220
1 × 10^5^	9/9	27.165	0.415
1 × 10^4^	9/9	30.611	0.353
1 × 10^3^	7/9	35.324	2.072
1 × 10^2^	2/9	36.665	1.831

## Data Availability

The original contributions presented in the study are included in the article, further inquiries can be directed to the corresponding author.

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
