# Peer review of "Development of a Sensitive and Specific Quantitative RT-qPCR Method for the Detection of Hepatitis E Virus Genotype 3 in Porcine Liver and Foodstuff"

_foods, 2024, doi:10.3390/foods13030467_

Round 1

Reviewer 1 Report

Comments and Suggestions for Authors

In the manuscript (ID foods-2843074), authors developed a sensitive and specific quantitative RT-qPCR method for the detection of hepatitis E virus genotype 3 in porcine liver and foodstuff. Overall, this topic has scientific significance and the manuscript has readability. Therefore, I think the following issue needs to be addressed:

(1) Title

--Line 25: Should be “Development of a Sensitive and Specific Quantitative RT-qPCR Method for the Detection of Hepatitis E Virus Genotype 3 in Porcine Liver and Foodstuff” rather “Development of a sensitive and specific quantitative RT-qPCR method for the detection of hepatitis E virus genotype 3 in porcine liver and foodstuff”.

(2) Abstract:

--Line 20: Please provide the full name of ORF1. When the full name appears in the preceding manuscript, the abbreviation is written after the full name, followed by the abbreviated form. This abbreviation can be written directly in the manuscript.

--Line 25: Should be “pig livers from different origins” rather “pig livers of different origins”.

(3) Keywords:

--Line 30: Should be “hepatitis E virus genotype 3” rather “hepatitis E virus”.

(4) Introduction: The introduction provides an adequate background on the topic.

-- The Introduction is too long, and it is suggested that the authors should shorten it appropriately.

--Line 58: high antibody und virus prevalence ?

--Line 58-61: In addition to high antibody und virus prevalence in the European domestic pig and wild boar population, HEV RNA has been detected in pork and products made from it at various stages of the food chain in a large number of studies. Please rewrite this sentence.

--Line 127: Should be “a highly sensitive quantitative RT-qPCR method” rather “a highly sensitive quantitative RT-qPCR”.

(5) Materials and methods

--Materials and reagents are omitted in this part. Suggest authors to add them. In addition, please check carefully and add information on relevant Materials and reagents. It is of utmost importance this is clarified and more detailed to allow replication.

--Line 132: Should be “2.1 Primer Design” rather “2.1 Primer design”. Please read the journal's requirements carefully and revise the title.

--Line 156: Should be “An IAC” rather “An internal amplification control (IAC)”.

(6) Results

-- Data analysis, especially significance analysis, should be applied to all experimental results of the manuscript, such as Figure 2.

Comments on the Quality of English Language

Minor editing of English language required.

Author Response

Dear reviewer,

thank you very much for your constructive comments and suggestions for improvement.

The authors have implemented these appropriately and hopefully to your satisfaction.

Comments and Suggestions for Authors

(1) Title

--Line 25: Should be “Development of a Sensitive and Specific Quantitative RT-qPCR Method for the Detection of Hepatitis E Virus Genotype 3 in Porcine Liver and Foodstuff” rather “Development of a sensitive and specific quantitative RT-qPCR method for the detection of hepatitis E virus genotype 3 in porcine liver and foodstuff”.

- Has been adjusted as suggested, see lines 1-4 on page 1.

(2) Abstract:

--Line 20: Please provide the full name of ORF1. When the full name appears in the preceding manuscript, the abbreviation is written after the full name, followed by the abbreviated form. This abbreviation can be written directly in the manuscript.

 - Has been adjusted as suggested, see lines 22-23 on page 1.

--Line 25: Should be “pig livers from different origins” rather “pig livers of different origins”.

- Has been adjusted as suggested, see line 28 on page 1

(3) Keywords:

--Line 30: Should be “hepatitis E virus genotype 3” rather “hepatitis E virus”.

- Has been adjusted as suggested, see line 33 on page 1.

(4) Introduction: The introduction provides an adequate background on the topic.

-- The Introduction is too long, and it is suggested that the authors should shorten it appropriately.

- Another reviewer suggested further elaboration of some aspects, so unfortunately a shortening is difficult to realise

--Line 58: high antibody und virus prevalence ?

- Has been rewritten as it was suggested, see line 61 on page 2.

--Line 58-61: In addition to high antibody und virus prevalence in the European domestic pig and wild boar population, HEV RNA has been detected in pork and products made from it at various stages of the food chain in a large number of studies. Please rewrite this sentence.

- Has been rewritten as suggested and formulated more precisely for better comprehensibility, see lines 59-64 on page 2.

--Line 127: Should be “a highly sensitive quantitative RT-qPCR method” rather “a highly sensitive quantitative RT-qPCR”.

- Has been adjusted as suggested, see line 133 on page 3.

(5) Materials and methods

--Materials and reagents are omitted in this part. Suggest authors to add them. In addition, please check carefully and add information on relevant Materials and reagents. It is of utmost importance this is clarified and more detailed to allow replication.

- Reagents and materials have been added, see lines 154-158 on page 4.

--Line 132: Should be “2.1 Primer Design” rather “2.1 Primer design”. Please read the journal's requirements carefully and revise the title.

- Has been adjusted as suggested, see line 138 on page 3.

--Line 156: Should be “An IAC” rather “An internal amplification control (IAC)”.

- Has been adjusted as suggested, see line 165 on page 4.

(6) Results

-- Data analysis, especially significance analysis, should be applied to all experimental results of the manuscript, such as Figure 2.

- The results were presented and described in text form for better readability, see lines 318 – 322 on page 9. Original results and further illustrations can be submitted as supplementary data if necessary.

Comments on the Quality of English Language

Minor editing of English language required.

- It was proofread again and edited when necessary.

Sincerely on behalf of the authors, Jan Hinrichs

Reviewer 2 Report

Comments and Suggestions for Authors

Jan Hinrichs et al. submitted to Foods an article dealing with the development of a sensitive and specific quantitative RT-qPCR method for the detection of HEV gen 3 in porcine liver and foodstuff.

The manuscript is well structured, but at current status it requires major additions to be adequately  implemented:

- Abstract section: before describing “the aim” of the study, briefly frame the setting in which your investigation fits;

-  introduction section: please implement the description relating to the food matrices involved in HEV outbreaks, in order to give a detailed focus adhering to food-borne diseases;

- discussions section: given the topic covered and the target Journal, it is necessary to integrate the discussions with detailed aspects relating to Public Health strategies, especially for healthcare workers involved in the operational lines of Food Safety, aimed at supporting the identification and management of cases of food-borne diseases attributable to HEV;

- discussions section: please better identify the strengths and future prospects of the study. 

Comments on the Quality of English Language

Minor editing of English language required

Author Response

Dear reviewer,

thank you very much for your constructive comments and suggestions for improvement.

The authors have implemented these appropriately and hopefully to your satisfaction.

Comments and Suggestions for Authors

Jan Hinrichs et al. submitted to Foods an article dealing with the development of a sensitive and specific quantitative RT-qPCR method for the detection of HEV gen 3 in porcine liver and foodstuff. The manuscript is well structured, but at current status it requires major additions to be adequately  implemented:

- Abstract section: before describing “the aim” of the study, briefly frame the setting in which your investigation fits;

- Thank you for the helpful comment. It has been adjusted as suggested, see line 17-19 on page 1.

-  introduction section: please implement the description relating to the food matrices involved in HEV outbreaks, in order to give a detailed focus adhering to food-borne diseases;

- Has been clarified as suggested and supplemented with additional references, see line 68-70 on page 2.

- discussions section: given the topic covered and the target Journal, it is necessary to integrate the discussions with detailed aspects relating to Public Health strategies, especially for healthcare workers involved in the operational lines of Food Safety, aimed at supporting the identification and management of cases of food-borne diseases attributable to HEV;

- Has been specified, see line 468-477 on page 13.

- discussions section: please better identify the strengths and future prospects of the study

- Has been emphasised, see line 468-477 on page 13.

Comments on the Quality of English Language

Minor editing of English language required.

- It was proofread again and edited when necessary.

Sincerely on behalf of the authors, Jan Hinrichs

Round 2

Reviewer 2 Report

Comments and Suggestions for Authors

The Authors correctly made improvements and modifications to the manuscript.

I would kindly ask to change "critical risk points" (L475) to "critical control points", in compliance with the provisions of the Community Regulations regarding HACCP.

Thank you.

Comments on the Quality of English Language

Minor editing of English language required